# Dnmt1a is essential for gene body methylation and the regulation of the zygotic genome in a wasp

**Deanna Arsala**[1,2], **Xin Wu**[3], **Soojin V. Yi**[3,4], **Jeremy A. Lynch**[1]*

1 Department of Biological Sciences, University of Illinois at Chicago, Chicago, Illinois, United States of America, 2 Department of Ecology & Evolution, University of Chicago, Chicago, Illinois, United States of America, 3 School of Biological Sciences, Georgia Institute of Technology, Atlanta, Georgia, United States of America, 4 Department of Ecology, Evolution and Marine Biology, University of California, Santa Barbara, Santa Barbara, California, United States of America

* jlynch42@uic.edu

**Data Availability Statement:** Raw data is available from the BioProject section at NCBI: WGBS data: https://www.ncbi.nlm.nih.gov/bioproject/PRJNA701143 RNA-seq data: https://www.ncbi.nlm.nih.gov/bioproject/PRJNA701367.

## Abstract

Gene body methylation (GBM) is an ancestral mode of DNA methylation whose role in development has been obscured by the more prominent roles of promoter and CpG island methylation. The wasp *Nasonia vitripennis* has little promoter and CpG island methylation, yet retains strong GBM, making it an excellent model for elucidating the roles of GBM. Here we show that *N. vitripennis* DNA methyltransferase 1a (*Nv-Dnmt1a*) knockdown leads to failures in cellularization and gastrulation of the embryo. Both of these disrupted events are hallmarks of the maternal-zygotic transition (MZT) in insects. Analysis of the embryonic transcriptome and methylome revealed strong reduction of GBM and widespread disruption of gene expression during embryogenesis after *Nv-Dnmt1a* knockdown. Strikingly, there was a strong correlation between loss of GBM and reduced gene expression in thousands of methylated loci, consistent with the hypothesis that GBM directly facilitates high levels of transcription. We propose that lower expression levels of methylated genes due to reduced GBM is the crucial direct effect of *Nv-Dnmt1* knockdown. Subsequently, the disruption of methylated genes leads to downstream dysregulation of the MZT, culminating in developmental failure at gastrulation.

## Author summary

The proper activation and regulation of the zygotic genome in the early stages of development is required for development to proceed past early mitotic divisions in insects. We have found that this process is profoundly disrupted after reduction of a DNA methyltransferase gene (*Nv-Dnmt1a*) in the wasp *Nasonia vitripennis*, resulting in failure of cellularization and morphogenetic movements. Like other insects, DNA methylation occurs almost exclusively in *Nasonia* gene bodies, and this methylation is strongly reduced after *Nv-dnmt1* knockdown. Expression levels from methylated genes are also reduced, in proportion to the degree of methylation loss. These results point to a key role of gene body methylation in regulating gene expression levels that are relevant to important

**Funding:** This work was supported by grant
MCB1615664 from the National Science
Foundation (nsf.gov) to SVY, and by grants
R01GM129153 and R03HD087476 from the
National Institutes of Health (nih.gov) to JAL. The
funders had no role in study design, data collection
and analysis, decision to publish, or preparation of
the manuscript.

**Competing interests:** The authors have declared
that no competing interests exist.

developmental events. This differs from studies in other insects, potentially indicating a
diversity of roles of gene body methylation and Dnmt1 orthologs among the insects.

## Introduction

The maternal-zygotic transition (MZT) is an essential stage in multicellular eukaryotic devel-
opment and represents the transition of developmental control from maternal genome prod-
ucts provided to the egg during oogenesis to the transcriptional products of newly formed
zygotic genome [1,2]. The MZT is regulated by many layers of pre- and post-transcriptional
mechanisms, including histone modification, chromatin packaging, miRNAs, mRNA decay
and degradation, and methylation of genomic DNA [1,2]. The interactions between the mater-
nal and zygotic genomes have been most comprehensively demonstrated in the insect model
*Drosophila melanogaster*. Maternal transcript destabilization, which is required for zygotic
genome activation (ZGA), is carried out in part by maternally deposited mRNAs encoding
RNA binding proteins (RBPs) [3,4] and aided by transcription factors such as Zelda (Zld),
which promotes miRNA expression that further destabilize maternal transcripts. Under nor-
mal circumstances Zld also increases chromatin accessibility and helps activate thousands of
zygotic genes to their proper level [5–7]. Disruption of either zygotic genome activation, or
maternal transcript clearance leads to embryonic lethality. In both cases, death is due to fail-
ures in the earliest developmental events that are dependent on the zygotic genome: cellulari-
zation of the syncytial blastoderm, and gastrulation [5,8].

Most of the strategies for regulating the *D. melanogaster* ZGA (e.g., histone modification,
chromatin packaging, action of miRNAs, mRNA degradation), and many of the molecules
involved, are conserved throughout the animals [9]. However, *D. melanogaster* lacks a major
class of genomic regulation, in the form of methylation of cytosines in CpG dinucleotides
(CpG methylation) [10,11]. CpG methylation can be catalyzed by DNA methyltransferases 1
and 3 orthologs (DNMT1 and DNMT3), is crucial for regulating gene expression in most com-
plex eukaryotes [12–14]), and has a significant role in regulating the MZT in many species
[15,16].

The most well understood functional impact of CpG methylation is in silencing gene
expression via methylation of cis-regulatory sequences in clusters of CpG nucleotides (CpG
islands) [13,17]. However, it is becoming clear that this is not a ubiquitous regulatory mecha-
nism [18]. A more universal, and likely ancestral [19,20], form of DNA methylation occurs
between the transcription start site and transcription end site of a gene, and is known as gene
body methylation (GBM) [21].

Previous studies have correlated GBM with less variable, and higher levels of gene expres-
sion than genes without GBM [20,22–25] or postulated that GBM might be a by-product of
open-chromatin states associated with active transcription [26], but its role in regulating gene
expression in specific developmental contexts such as the MZT is poorly understood. Whole-
genome bisulfite sequencing (WGBS) studies have shown that DNA methylation patterns are
dynamic during the MZT in vertebrate model systems [27–29]. However, since manipulating
DNA methylation machinery in vertebrate models affects both CpG islands and GBM, it is dif-
ficult to identify functions of GBM that are independent of CpG island methylation.

In contrast to vertebrates, DNA methylation occurs almost exclusively at gene bodies in
insects [30–33]. Thus, such insect model systems have the potential to make significant contri-
butions in understanding the role of GBM in development. The wasp *Nasonia vitripennis* is a
particularly attractive system with a full methylation toolkit, harboring orthologs of the

maintenance DNA methyltransferase, DNMT1, and the *de novo* DNA methyltransferase, DNMT3, both of which exclusively mediate gene body methylation [31,34,35].

There are three DNMT1 paralogs and one DNMT3 ortholog in the *N. vitripennis* genome, and one of them, *Nv-DNA methyltransferase 1a* (*Nv-Dnmt1a*), was shown to be essential for embryogenesis, while knockdown of the other two DNMT1 paralogs and DNMT3 ortholog produced viable embryos [36]. Therefore, *N. vitripennis* offers an exciting opportunity to functionally study the specific role of gene body methylation in a well-defined developmental system. In this work, we set out to experimentally investigate the role of *Nv-Dnmt1a* and infer the role for gene body methylation in early embryogenesis of *N. vitripennis*.

Our detailed analysis of effects of *Nv-Dnmt1a* depletion on early embryogenesis revealed major defects in cellularization and morphogenesis of the early embryo, which are phenotypes typically seen after disruption of the MZT in insects. Using parental RNAi (pRNAi) and WGBS we found that the vast majority of gene body methylation is lost when *Nv-Dnmt1a* is knocked down. We then show that loss of gene body methylation is significantly correlated with reduced gene expression levels from methylated loci during zygotic genome activation. We propose that *Nv-Dnmt1a* is required for maintaining methylation at gene bodies and disruption of methylated gene expression after *Nv-Dnmt1a* knockdown has cascading effects that dysregulate thousands of genes during the MZT, leading to failure in the first developmental processes that rely on a properly activated zygotic genome.

## Results

### Nv-Dnmt1a function is required for early embryonic events that require proper regulation of the zygotic genome

The initial characterization of *Nv-Dnmt1a* showed that knockdown through parental RNAi (pRNAi), where female embryos are injected with dsRNA [37], led to lethality of embryos produced by injected females [36]. To gain more insight into the specific functional importance of *Nv-Dnmt1a* for early embryonic development, we examined the embryonic developmental failures following *Nv-Dnmt1a* pRNAi in more detail.

Early development is very rapid in *N. vitripennis*. Nuclei divide rapidly, simultaneously, and without the formation of cell membranes to fill the large, unicellular egg with syncytial nuclei. The first 7 division cycles occur deep in the yolk, after which nuclei migrate to the cortex of the embryo, forming a syncytial blastoderm [38] consisting of a single layer of nuclei populating the entire cortex of the egg. Syncytial blastoderm nuclei divide five more times, prior to cellularization. We observed no differences between control embryos and *Nv-Dnmt1a* pRNAi embryos during the first 11 syncytial divisions (Fig 1A).

The first obvious defect occurred at the 12th division cycle (Fig 1B and Fig 1D). Normally, nuclei become cellularized by encroaching plasma membrane, and immediately begin the morphogenetic movements of gastrulation at the end of this stage [38]. The first evidence of morphogenetic movements in control embryos was observed 193 minutes after the onset of nuclear cycle 12, on average (n = 5, SD = 13.12, (Fig 1A)), similar to our previously published results in untreated embryos [39]. However, apparent morphogenetic movements in *Nv-Dnmt1a* pRNAi embryos began significantly earlier (171 minutes (n = 5, SD = 5.98) into cycle 12, on average) (Fig 1A).

In addition to being premature, these movements were abnormal. In *Nv-Dnmt1a* pRNAi embryos, a line of cells at the anterior end of the embryo began to ingress posteriorly (Fig 1D) which was not seen in control embryos (Fig 1B). In addition, internalization of the mesoderm and migration of the serosa from the dorsal pole to cover the embryo both failed (Fig 1C and Fig 1E, see Buchta et al., for more description of these movements [38]).

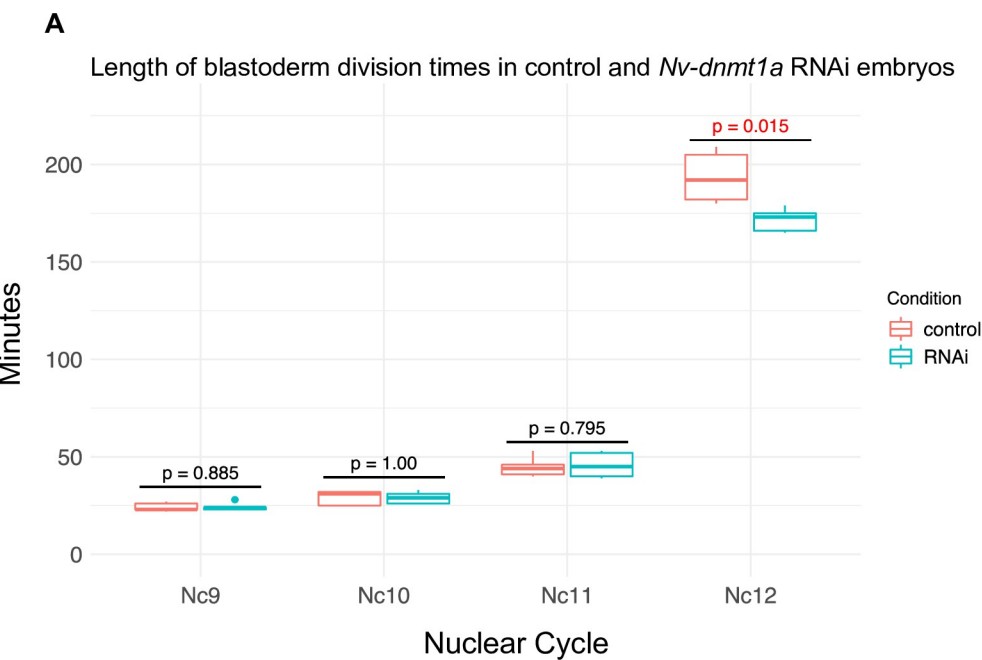

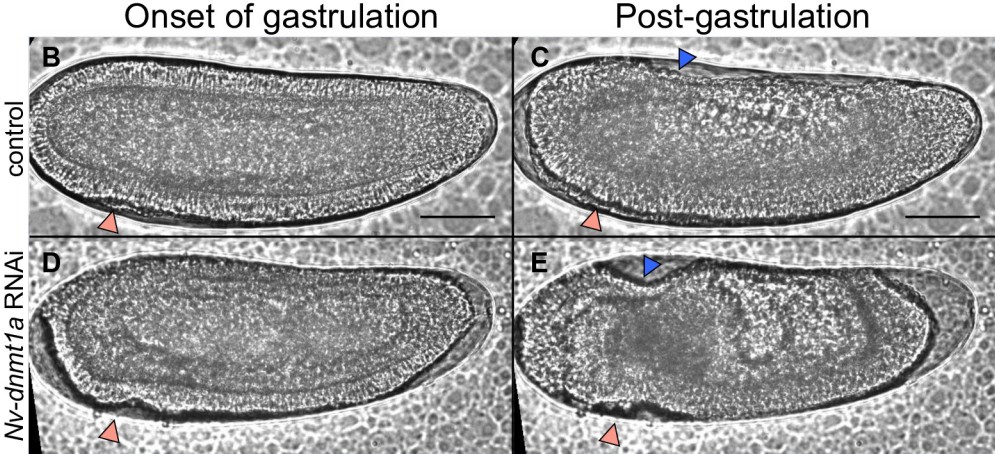

**Fig 1. Live imaging of control and *Nv-dnmt1a* RNAi embryos.** A) Box and whisker plot displaying the length of blastoderm division times that were measured during live imaging experiments. Nc = nuclear cycle #. P values were determined by performing a 2-tailed T-test assuming unequal variance between RNAi and control. N = 5 control, N = 5 Nv-dnmt1a RNAi. B-C) Live-imaging of a control embryo at the onset of gastrulation and at post-gastrulation. Peach arrows in (B) point to the initial mesodermal folds which are eventually internalized C. The blue arrowhead in C points to the serosa. Scale bars (black lines in B and C) are 50μm. (D-E) Live-imaging of a Nv-dnmt1a RNAi embryo at the onset of gastrulation and at what is equivalent to post-gastrulation. Defective mesodermal folds are denoted by peach arrowheads in D and E. Abnormal serosal movements are denoted by a blue arrowhead in E.

To get a more detailed understanding of the cellular basis of these defects, we examined high-resolution images of transverse sections *Nv-Dnmt1a* pRNAi and control embryos at the end of the 12th cell cycle, when the cellularization and the first gastrulation movements should be occurring.

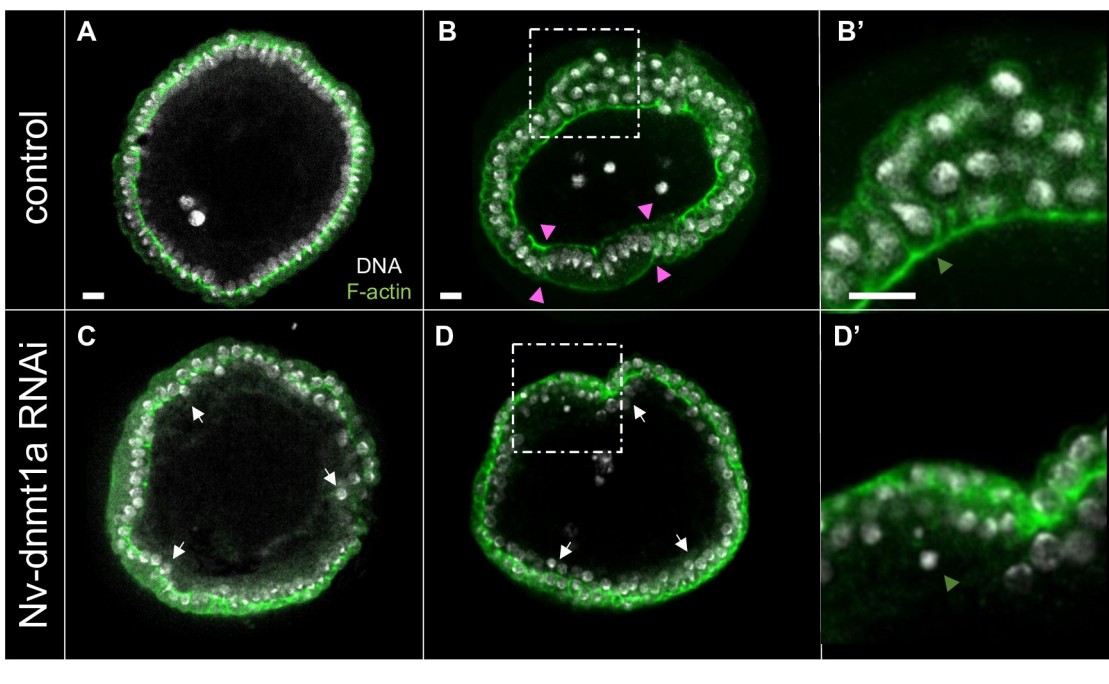

**Fig 2. Transverse views of control and *Nv-dnmt1a* pRNAi embryos during cellularization and at the onset of gastrulation.**
Green, phalloidin (F-actin). white, DAPI (DNA). (A) Control embryo during mid cellularization. (B) Control embryo at the onset of gastrulation. Pink arrowheads demarcate the borders of the presumptive mesoderm. (B') Zoomed in view of fully cellularized control embryo. Orange arrowhead points to the basal actin cable. (C) *Nv-dnmt1a* pRNAi embryo during mid-cellularization. White arrows point to examples of disordered nuclei. (D) *Nv-dnmt1a* pRNAi embryo at the onset of gastrulation. White arrows point to examples of non-cellularized, disordered nuclei. (D') Zoomed in view of an *Nv-dnmt1a* pRNAi embryo that fails to fully cellularize. Orange arrowhead points to a non-cellularized nucleus (note the absence of the basal actin cable). (E) Table with n values of cellularized and non-cellularized control and *Nv-dnmt1a* pRNAi embryos. *** = p-value < 0.0001 determined by Fisher's Exact Test. All scale bars (white lines in A-A''') represent 10 μm.

All control embryos in the process of cellularization, displayed membrane ingression that was consistent around the entire blastoderm circumference (Fig 2A). In contrast, membrane ingression was uneven in *Nv-Dnmt1a* pRNAi embryos, leaving many nuclei completely or mostly unencapsulated prior to gastrulation (Fig 2C, white arrows point to examples of non-cellularized nuclei).

All (24/24) control embryos initiating morphogenesis had nuclei that were fully enveloped by the plasm membrane and had an actin cable along the basal (interior) surface of the cellular blastoderm (Fig 2B–2B', orange arrowhead in 2B' marks the basal actin cable). Enlarged, round serosal cells were present on the dorsal side, and the distinct population of presumptive mesodermal cells was present on the ventral side (Fig 2C, pink arrowheads mark the borders of the presumptive mesoderm). In contrast, *Nv-Dnmt1a* pRNAi embryos had many unencapsulated nuclei, and lacked the cable of actin along the interior surface of the blastoderm (Fig 2D–2D', orange arrowhead marks non-cellularized nuclei and obvious absence of the basal actin cable), consistent with failed cellularization (23 out of 24 pRNAi embryos were

incompletely cellularized). In addition, the blastoderm was multilayered at many locations, and clearly distinct serosal and mesodermal precursors were not observed (Fig 2D).

## Nv-Dnmt1a is required for global gene body methylation

The failure of cellularization and morphogenesis are widely conserved indicators of disrupted MZT [39–41]. We hypothesized that *Nv-Dnmt1a* is required for proper regulation of the MZT, and its reduction by pRNAi would abrogate DNA methylation and consequently disrupt zygotic genome activation in the early embryo of *N. vitripennis*. We thus sought to characterize the DNA methylation patterns in the early embryo of *N. vitripennis* and to examine how this pattern was affected when *Nv-Dnmt1a* was depleted by pRNAi.

In control embryos collected at the late blastoderm stage (7–9 hours after egg-lay) we identified 151,480 methylated CpGs (mCGs) in a union set representing all of the different mCGs across three biological replicates. Each sample contained similar numbers of mCGs across the three biological replicates (ranging from 136,410 to 136,717). The vast majority of these mCGs were found within gene bodies (see Table 1 for summary CpG statistics).

Fractional DNA methylation of gene bodies shows the canonical bimodal distribution (shown for a representative sample, Fig 3A) as seen in other invertebrate species including insects [21]. Following the criteria used in a previous study, we designated genes with gene body fractional methylation levels of at least 0.02 as methylated genes [21]. By this criterion we found 5,423 methylated genes in the blastoderm embryo, where 99% of our methylated genes in the embryo overlapped with methylated genes identified in adult *N. vitripennis* tissues [31], where a slightly different method to define methylated genes was used (S1 Data). This indicates that our results are highly compatible with those of Wang et al. [31], and are consistent with the idea that GBM patterns are generally stable across life stages.

We knocked-down *Nv-Dnmt1a* mRNA with pRNAi and examined DNA methylation at the same developmental stage (7–9 hours after egg lay) as the control. Based on our RNA-seq results, *Nv-Dnmt1a* mRNA levels were reduced by 5-fold or more in late blastoderm pRNAi samples relative to late blastoderm control samples (S1 Data and S2 Fig). We observed a significant reduction in DNA methylation at 99.7% of methylated CpGs in the genome in pRNAi embryos. The magnitude of the reduction was substantial, as we found an average of 81.5% DNA methylation depletion per mCG (Fig 3B and 3C and Table 1).

While DNA methylation was strongly reduced genome-wide, the intragenic pattern of GBM was maintained. For example, mCGs were enriched in exons and were more numerous at the 5' end of the coding region of gene bodies in control embryos (Fig 3D) similar to patterns found in other insects and other developmental stages [30,42,43]. This pattern was maintained in pRNAi embryos, despite the drastic overall reduction in DNA methylation levels (Fig 3D). This suggests that *Nv-Dnmt1a* is required for maintenance of gene body methylation but does not govern the 5' and exon-biased pattern of DNA methylation that we see in *N. vitripennis* and other hymenopteran insects.

**Table 1. Summary CpG statistics and non-conversion rates for each sample.**

| Sample | Non-conversion rate | Median CpG coverage | Mean CpG coverage | Methylated CpGs (mCGs) | mCGs within gene bodies |
|---|---|---|---|---|---|
| Control 1 | 0.0052 | 14 | 14.92 | 136,600 | 126,333 |
| Control 2 | 0.0051 | 15 | 15.63 | 136,410 | 126,174 |
| Control 3 | 0.0055 | 14 | 14.75 | 136,717 | 126,364 |
| RNAi 1 | 0.0052 | 14 | 14.3 | 98,511 | 91,874 |
| RNAi 2 | 0.0051 | 14 | 14.48 | 8,017 | 7,720 |
| RNAi 3 | 0.0053 | 14 | 14.91 | 15,305 | 14,669 |

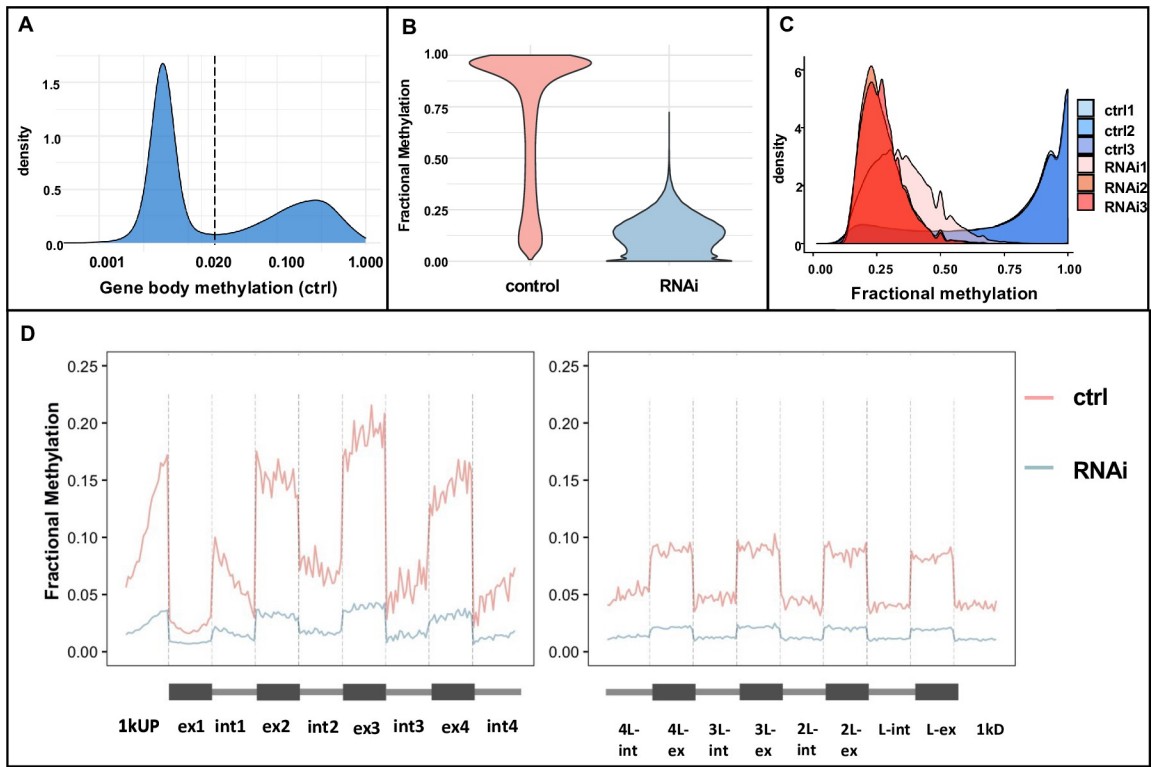

**Fig 3. Global DNA methylation in control and *Nv-dnmt1a* pRNAi late blastoderm *N. vitripennis* embryos.** (A) An example of distribution of fractional DNA methylation in a control sample exhibiting the classical bimodal distribution. We defined methylated genes as genes with a minimum of 0.02 gene body methylation (shown as the dashed line) to classify methylated and unmethylated genes. (B) Violin plot demonstrating the reduction of whole genome DNA methylation in *Nv-dnmt1a* pRNAi samples compared to the control samples. We calculated the average fractional methylation of all methylated CpGs in the control samples and corresponding positions in the pRNAi samples (~151k methylated CpGs). (C) Distributions of fractional methylation of CpGs that are methylated at least one control sample demonstrate that CpGs lose DNA methylation in RNAi embryos. (D) The pattern of mean fractional methylation in the first and last four exons and introns of all methylated genes in control and pRNAi samples, divided into equal-sized bins. Control shown in red, pRNAi shown in blue.

## Loss of Nv-Dnmt1a function disrupts the regulation of a large proportion of the early embryonic transcriptome

Given the profound effects of *Nv-Dnmt1a* on global DNA methylation and early developmental events, we sought to understand the effects of *Nv-Dnmt1a* knockdown on gene expression over time in the early embryo. Although the developmental effects *Nv-Dnmt1a* pRNAi were not visible until cellularization and the onset of morphogenesis, we hypothesized that changes in the transcriptome in response to disrupted development would occur earlier in development, leading up to the multifaceted failure of development at gastrulation.

To test this, we performed RNA-seq on timed egg collections from mock injected (control) and *Nv-Dnmt1a* dsRNA injected wasps at time points that cover early embryogenesis from egg laying to gastrulation (S1 Fig). The collection time points were: freshly laid eggs (0–2 hours after egg lay), early blastoderm (3–5 hours after egg lay), middle blastoderm (5–7 hours after egg lay) and late blastoderm/onset of gastrulation (7–9 hours after egg lay).

To confirm knockdown, we found that *Nv-Dnmt1a* was strongly down-regulated in pRNAi samples in all stages (S2 Fig and S2 Data). To asses specificity of the knockdown, we examined the expression of *Dnmt1* paralogs *Nv-Dnmt1c* and *b*, as well as the related gene *Nv-Dnmt3*. *Nv-Dnmt1c and Nv-Dnmt3* were unaffected after *Nv-Dnmt1a* pRNAi in all stages (S2 Fig and S2

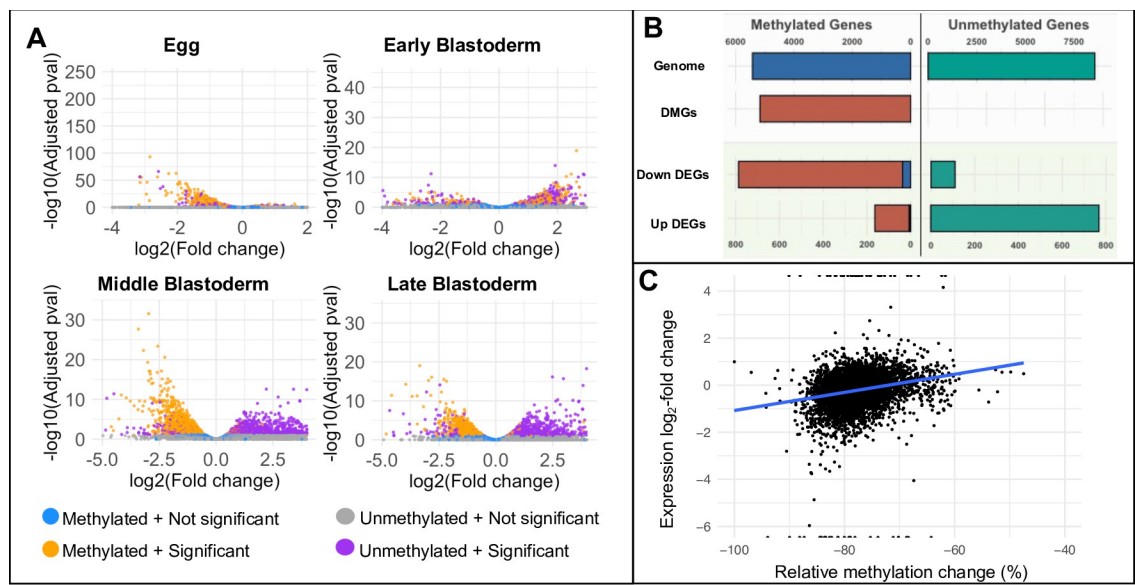

**Fig 4. Gene expression of methylated and unmethylated genes in control and *Nv-dnmt1a* pRNAi *N. vitripennis* embryos.** (A) Volcano plots showing differentially expressed genes between control and *Nv-dnmt1a* pRNAi in embryos of different developmental stages. X axis is log2 fold change in *Nv-dnmt1a* pRNAi samples relative to control samples. Y axis is log10 adjusted p-values. Golden dots are methylated, significantly differentially expressed (FDR-corrected p-value < 0.1). Purple dots are significant and DEGs that are unmethylated (FDR-corrected p-value < 0.1). Blue and grey are non-significant and represent methylated and unmethylated genes, respectively. (B) Distribution of methylated and unmethylated genes for all genes in the genome, differentially methylated genes (DMGs), down-regulated differentially expressed genes (Down DEGs), and up-regulated differentially expressed genes (Up DEGs). (C) The correlation between mean relative GBM change and mean relative expression change (calculated as log2-fold change) in the pRNAi samples compared to the control samples was positive and statistically significant (Spearman's rank correlation coefficient = 0.37, p-value < 2.2e-16).

Data). *Nv-Dnmt1b* expression was significantly reduced in the middle blastoderm stage after *Nv-Dnmt1a* pRNAi. Since, in contrast to *Nv-Dnmt1a*, it is not affected in the early stages, we do not believe this effect is due to cross reaction with our *Nv-Dnmt1a* dsRNA fragments. Rather, it is likely that the *Nv-Dnmt1b* locus is a downstream target of *Nv-Dnmta*, as *Nv-Dnmt1b* is a methylated gene in *Nasonia* (S2 Fig and S2 Data).

We found 835 and 888 differentially expressed genes (DEGs) in the egg and early blastoderm stages, respectively. The affected loci at these stages likely reflect the effects of the knockdown in the maternal nurse cells, as there is little to no zygotic transcription at these stages. Interestingly, 95% of all methylated genes in the *N. vitripennis* genome were represented in the maternal transcript pool. However, the knockdown effects were mild in both the number of genes affected and the magnitude of the effects compared to the following two stages (Fig 4A, S3 Table and S3 Fig). In the middle-blastoderm stage, where the zygotic genome is first broadly activated and maternal mRNAs are being cleared, 2744 genes showed significant differential expression, while in the late blastoderm stage 1831 were significantly differentially expressed.

Aside from the massive jump in the number differentially expressed genes at the mid-blastoderm stage, other intriguing patterns emerge at this time. Roughly equal numbers of the DEGs were up-regulated as were down-regulated. However, there was a clear bias in the methylation status of the DEGs and their direction of expression change. For example, 1201 out of 1331 (90%) of the down-regulated transcripts were derived from methylated genes in the middle blastoderm stage, while 1227/1413 (87%) upregulated transcripts came from unmethylated genes (Fig 4A, S3 Table and S3 Fig). Both observations were statistically significant (Fisher's

exact test, P < 0.001). A nearly identical pattern was maintained into the late blastoderm stage (Fig 4A and 4B, S3 Table, and S3 Fig).

The strong over-representation of methylated genes among down-regulated genes following *Nv-Dnmt1a* pRNAi fits well with previous observations in insects connecting DNA methylation with high and stable gene expression [31,33,44]. Furthermore, we have found that the magnitude of the reduction in expression was significantly correlated with the degree of lost methylation after pRNAi (Fig 4C, Spearman's rank correlation coefficient = 0.37, P < 2.2e-16).

These observations imply that GBM plays an important role in maintaining normal levels of high expression from methylated loci, and its loss leads to widespread reduction in transcription from these loci. Assuming that GBM is the major direct output of *Nv-Dnmt1a* function, the effects on unmethylated DEGs (uDEGs), and potentially some methylated DEGs, may be due to indirect effects downstream of the initial reduction of methylated DEGs (mDEGs) after pRNAi.

While indirect effects on unmethylated genes could be predicted, the very strong bias toward upregulation of this gene class was quite surprising. We compared our raw RNA-seq data and DESeq2 normalized data, to investigate if any technical bias due to the normalization method could have affected our inference of gene expression changes. There were no concerning or obvious differences between the distribution of raw read counts and the normalized counts used for our analyses (S4 and S5 Figs). Further, the size-factors required for normalization were all relatively small (S1 Table).

## Methylated and unmethylated genes show stark differences in their regulation during maternal-zygotic transition

We further examined all methylated and unmethylated genes over developmental time, focusing on those that showed differential expression between control and *Nv-Dnmt1a* pRNAi treatments in at least one stage (a total of 3904 genes). Among these, we identified 1,769 methylated differentially expressed genes. Plotting the expression levels of the mDEGs revealed consistent unimodal distributions with peaks centered near the median expression level in each stage (Fig 5). A similar pattern is seen when all methylated genes are plotted (S6 Fig). *Nv-Dnmt1a* pRNAi did not affect the shape of these distributions, but rather resulted in a modest, but significant shift of mDEG expression distributions downward during the middle and late blastoderm stages (Fig 5). This is consistent with our hypothesis that *Nv-Dnmt1a* dependent GBM plays a role in aiding the efficiency of transcription of methylated genes.

We identified a total 2135 uDEGs. The distribution of expression levels for these genes was more complicated, showing a bimodal tendency, with a peak at very low expression levels during the egg through middle blastoderm stages (Fig 5A, 5B and 5C), and less pronounced peaks at higher levels that did not overlap the medians. *Nv-Dnmt1a* pRNAi had little effect on global distribution of uDEG expression levels in the egg and early blastoderm stages (Fig 5A and 5B), consistent with the minor effects of pRNAi on these stages reported earlier (Fig 4A). However, in the middle and late blastoderm stages (Fig 5C and 5D), the median uDEG expression increases slightly in pRNAi samples compared to control embryos, and there was a pronounced reduction of the lowest expression levels (below 2.5). This indicates that uDEGs that were effectively absent in control embryos were present at meaningful levels after *Nv-Dnmt1a* pRNAi.

To understand how the MZT is impacted by *Nv-Dnmt1a* knockdown and subsequent GBM loss, we first examined the behavior of genes whose expression levels changed greater than two-fold in either direction between time points. These are potentially targets of the genome activation aspect of the MZT (genes that increase significantly from one time point to the

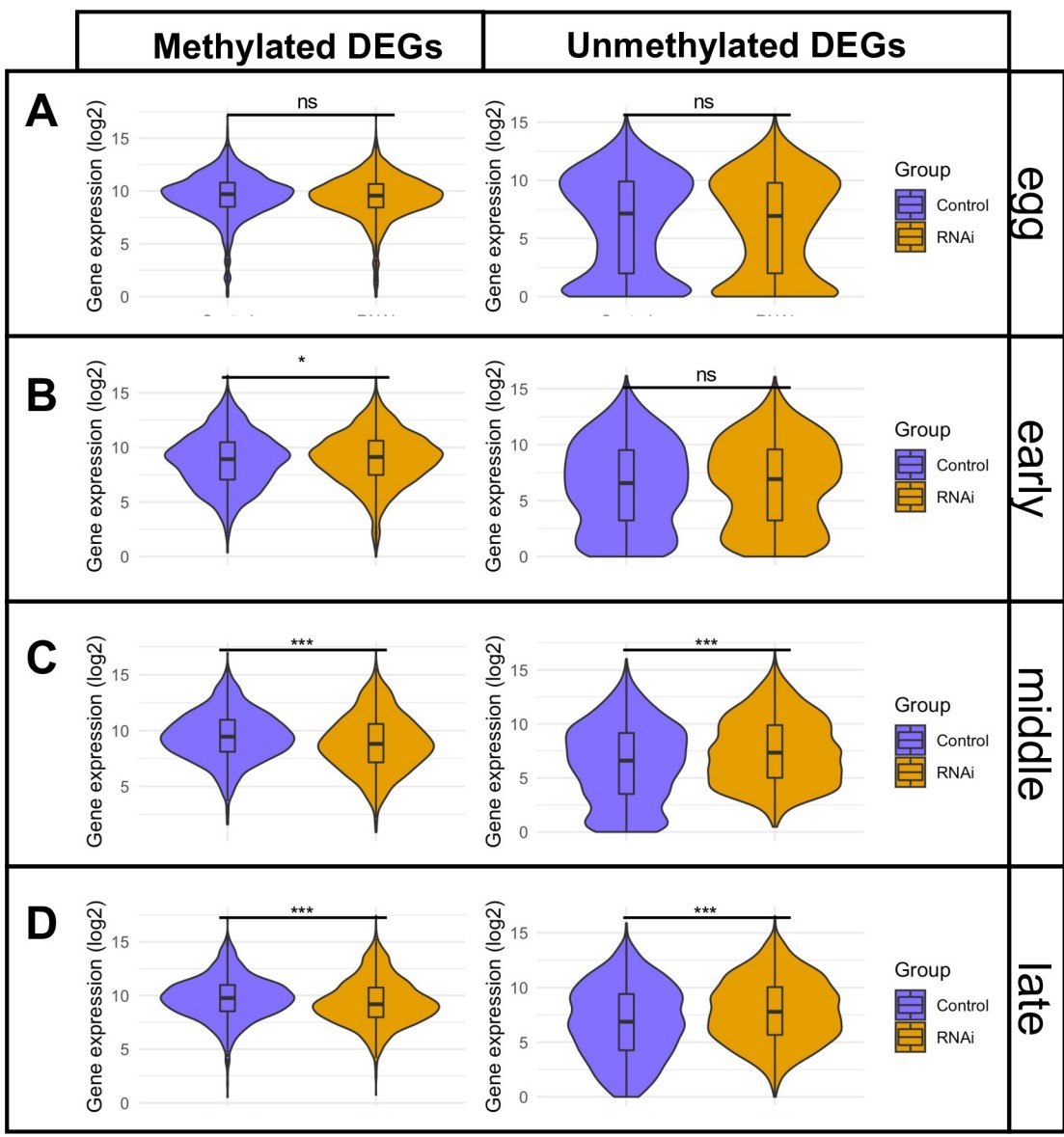

**Fig 5. Violin plots of methylated and unmethylated differentially expressed genes (DEGs) in control and *Nv-dnmt1a* pRNAi samples.** (A) eggs, (B) early blastoderm embryos, (C) middle blastoderm embryos, and (D) late blastoderm embryos. p-values were determined by performing a 2-tailed T-test assuming unequal variance between RNAi and control. ns = not significant (P>0.05). * = P<0.05. *** = P <0.0001.

next) or mRNA degradation (those that decrease significantly from on time point to the next). We then compared this to the pattern after *Nv-Dnmt1a* knockdown.

Despite the general characterization of methylated genes as "stably expressed" [20,22–25], many methylated genes change significantly throughout MZT during early embryogenesis. At each transition point, nearly a thousand or more methylated genes go up or down greater than 2-fold (S7 Fig). We also found that there was a surge of down-regulation in the transition from egg to early blastoderm, likely correlating with degradation of maternal mRNA of about 2000 genes (S7 Fig). In addition, we found that largest number of genes are upregulated between

timepoints was during the transition from early to middle blastoderm stages, indicating that this time period corresponds to a significant leap in *N. vitripennis* ZGA.

When we compared the set of genes whose expression levels changed over time in pRNAi embryos to the corresponding set in control embryos, the most significant divergences were observed in the transition for early to middle blastoderm stage (S7 Fig), further indicating that this is a crucial time point for MZT regulation. For example, there were 1758 transcripts derived from methylated genes that do not change levels significantly between early and middle blastoderm stages in control, but whose levels decreased by greater than two-fold across these timepoints after pRNAi (S7 Fig). This set represents 17% of the total number of genes annotated in the *N. vitripennis* transcriptome, and nearly a third of all the methylated genes. Reciprocally, over 1000 unmethylated genes that do not change from early to middle blastoderm in control show greater than two-fold up-regulation in the mid blastoderm stage, relative to the early blastoderm in *Nv-Dnmt1a* pRNAi embryos. This suggests that Nv-Dnmt1a function has an important, if likely indirect, role in negatively regulating these genes during the MZT. Both anomalies at the early-to-middle blastoderm transition are consistent with our general observations that mDEGs are mostly downregulated, while uDEGs are primarily upregulated.

To get further perspective on the patterns of expression change across the MZT, we used hierarchical clustering of gene expression across the four developmental timepoints to identify patterns of change across development and between control and knockdown embryos. With this approach we identified a cluster of mDEG transcripts that is unusual in that the transcripts show upregulation in the pRNAi samples relative to control at the early blastoderm stage, while the vast majority of differentially expressed methylated genes in our analysis are downregulated after pRNAi (for example see cluster outlined in green in S8A Fig). The transcripts in this cluster are maternally deposited at high levels, that then fall significantly in the early blastoderm stage in both control and pRNAi samples. Expression levels of these transcripts decrease precipitously in the subsequent early blastoderm stage in control, while they decrease to a lesser degree in control samples (S8A Fig, blue box). Further analysis showed that the levels of these transcripts are statistically indistinguishable between the two conditions in the egg but are significantly higher in the knockdown samples in the early blastoderm stage (S8B Fig), strongly suggesting that these genes are less efficiently degraded in the *Nv-dnmt1a* embryos in the early stages of the MZT. A similar cluster can be found among the uDEGs (blue box in S8C Fig), suggesting regulation of mRNA decay also affects unmethylated gene expression patterns after *Nv-Dnmt1a* knockdown.

We further identified two clusters uDEGs whose temporal expression dynamics are affected after pRNAi. First, we observed a cluster of uDEGs that show a significant general increase in pRNAi middle and late blastoderm stages, while they remain mostly unchanged and at low levels in the control (S8C Fig, black box, S8D Fig, S2 and S3 Data), suggesting improper activation of these genes after pRNAi.

We observed a second set of uDEGs that are activated zygotically during the early blastoderm stage, whose levels decrease during the middle blastoderm stage in control samples but does not occur in pRNAi samples (S8C Fig, see red dotted box). This suggests that regulation of mRNA degradation is affected downstream of *Nv-Dnmt1*a pRNAi during the major phase of the MZT, along with the previously observed effect in the egg to embryo transition (S8A and S8C Fig, blue boxes).

While there are likely many complex interactions that lead to differential expression after *Nv-dnmt1a* pRNAi, these clusters show more clearly that both regulation of mRNA stability, and repressive regulation of gene expression are disrupted when *Nv-Dnm1a* function is reduced. Since GBM is associated with increased gene expression, and these disruptions served

to increase expression of unmethylated genes, we hypothesize that these effects come about indirectly through methylated, direct, targets of *Nv-Dnmt1a*.

## A complex relationship between the targets of *Nv-Dnmt1a* and its pRNAi phenotype

We examined the annotations of the methylated DEGs and identified many differentially expressed factors, such as RNA binding and regulating factors (including the well-known MZT factors *Smaug* and multiple subunits of the CCR4-NOT deadenlyase complex), basal transcription factors, cyclins and cytoskeletal components that could plausibly play important roles in producing the observed patterns of gene expression change and developmental failure after *Nv-Dnmt1a* pRNAi (S4 Data). Given the similarity of the developmental phenotypes of *Nv-dnmt1a* to those where the MZT was disrupted, it is potentially significant that the MZT component *Nv-smaug*, was significantly up-regulated during the early blastoderm stage after pRNAi (it is an example of the cluster of maternal transcripts that are not fully degraded in the transition from egg to early blastoderm samples (S8A Fig, red box). However, beyond this extended persistence of the maternal RNA in the early blastoderm, levels returned to normal in the later stage of RNAi (S9 Fig).

We also examined expression and methylation status of *Nv-zelda*, because pRNAi against this gene phenocopies the developmental defects of *Nv-Dnmt1a* [39]. However, *Nv-zelda* is neither methylated, nor is its expression affected by *Nv-Dnmt1a* knockdown (S9 Fig). This suggests that *Nv-dnmt1a* does not act through *Nv-zelda* to regulate cellularization and gastrulation.

We next examined the effects of *Nv-Dnmt1a* knockdown on expression of the remaining candidate mDEGs with potentially relevant functional annotations in the middle blastoderm stage (S4 Data). The changes were relatively moderate, neither completely reducing or majorly increasing in levels after *Nv-Dnmt1a* knockdown, so neither loss of function nor gain of function experiments would likely produce phenocopies of the *Nv-Dnmt1a* pRNAi. Rather, the observed patterns are consistent with moderate disruption of several genes interacting to produce the complex cellularization and morphogenesis phenotypes.

Gene ontology analysis found that terms involved in metabolism, and other basal cell functions were the most significantly and consistently enriched in *Nv-dnmt1a* target genes (down-regulated, methylated genes) in the middle and late blastoderm stages (S2 Table). Interestingly, RNA degradation was the most significantly enriched term among *Nv-dnmt1a* target genes, (S2 Table), consistent with our RNA-seq results.

## Discussion

A previous study found that only one of the DNMT1 orthologs in the wasp (*Nv-Dnmt1a*) was required for *N. vitripennis* embryogenesis, while the other two DNMT1 paralogs (*Nv-Dnmt1b*, *Nv-Dnmt1c*) and the DNMT3 ortholog *Nv-dnmt3* were dispensable for embryonic development [36]. While the sequences of the Nv-Dnmt paralogs are significantly similar across their length, overall amino acid identity among ranges from only 45–65%. This divergence leaves plenty of room for specialization and sub-functionalization of the paralogs. This may have led to *Nv-Dnmt1a* to become the main indispensable factor in regulating early development, and leave open the possibility that the other two paralogs are specialized for different time points and developmental contexts. *Nv-Dnmt1a* pRNAi embryos failed to form segments and the knockdown was lethal [36] which led us to examine the phenotype in more detail.

We found that the first signs of developmental failure are nuclei that fail to be correctly cellularized. This directly precedes failure of the morphogenetic processes of mesoderm

internalization, and migration of the serosa. These are the first developmental events that are dependent on the zygotic genome in insects and are the developmental events that are strongly affected when the MZT is disrupted in *N. vitripennis* and *D. melanogaster*. We hypothesized that *Nv-Dnmt1a* is required for GBM and the regulation of gene expression during the MZT. Consistent with this, *Nv-Dnmt1a* knockdown resulted in major disruption of gene expression levels during early embryogenesis. The most significant alterations to the transcriptome occurred at the mid blastoderm stage, which correlates well with when the major wave of zygotic genome activation occurs in *N. vitripennis*. This observation, in combination with our finding that GBM was significantly reduced after *Nv-Dnmt1a* pRNAi, strongly suggests that loss of gene body methylation disrupts the MZT in *N. vitripennis*.

A potential caveat to our interpretation is that we knock down *Nv-Dnmt1a* maternally, allowing the possibility that effects on maternal genes could be the ultimate cause of the later phenotypes. Since there are genes that show differential expression in the egg, prior to the onset of zygotic gene expression, we cannot formally exclude this possibility. However, the number of these genes is low, and the magnitude of the effect of pRNAi in the egg is small and transient (Figs 4A and 5A), as most of these gene return to near-control levels later in development (S5 Data). In addition, a cascading set of indirect effects would not likely lead to the pattern of gene expression changes we see, where mDGEs are nearly universally downregulated, while uDEGs are similarly universally upregulated relative to control.

The exact mechanism by which *Nv-Dnmt1a* regulates the MZT and downstream processes is not yet understood. However, a crucial observation was that there was a very strong correlation between gene body methylation state and the direction of expression change after *Nv-Dnmt1a* pRNAi: downregulated genes in pRNAi embryos were almost exclusively methylated genes (which were confirmed to have reduced GBM), and upregulated genes were conversely almost exclusively unmethylated (Fig 4).

The correlation of reduced GBM with reduced gene expression among methylated genes after *Nv-Dnmt1a* pRNAi is consistent with previous observations that GBM is associated with high, steady levels of gene expression [31,45]. However, it has not been conclusively shown that GBM directly increases levels of gene expression, and the exact role played by GBM has been controversial [46,47]. Comparison of orthologous gene expression between plant species or populations that had either maintained or lost GBM capability revealed little or no effect of GBM on expression levels of ancestrally methylated genes, indicating that GBM may have no appreciable effect on transcription [46–48]. In insects, knockdown of DNMT1 orthologs lead to the loss GBM in the roach *Blatella* and the milkweed bug *Oncopeltus* [30,33]. In the roach, loss of DNMT1 led to disruption of methylated gene expression levels, but no bias toward reduced expression was observed, while in the milkweed bug no change in gene expression levels was observed for any genes [30,33].

The above results appear to conflict with our findings that, after *Nv-Dnmt1a* pRNAi, the vast majority of methylated genes that are differentially expressed show reduced expression, and that the magnitude of reduced expression directly correlated with the degree of gene body methylation loss. This may indicate that the role of GBM methylation is quite labile in evolution, and its significance and exact role might vary from species to species. We also propose that our highly focused analysis of early embryogenesis is advantageous for detecting moderate and transient effects of altered GBM. The pre-gastrulation embryo is a relatively homogenous tissue, where cells have not completed differentiation. Given that we are examining the very early stages of embryo development, we might be able to capture early response to the loss of GBM without being obscured by noise introduced by indirect effects and expanded cellular heterogeneity in later stages. In the other studies, gene expression was examined in complex tissues that were developed well beyond the earliest potential effects of GBM loss. We believe

further taxonomic sampling that focuses on time limited and homogenous tissues will reveal the relative importance of these ideas.

Another complication for our interpretation of *Nv-Dnmt1a's* function is evidence that DNMT1 orthologs have roles independent of their methylation functions in other insects. While reduction of *dnmt1* in *O. fasciatus* reduces methylation genome-wide, and results in sterile ovarioles and testes no apparent effect on gene expression was detected in ovaries suggesting that GBM was not the important function of *Of-Dnmt1* at this stage [30,49]. In *T. castaneum*, GBM is absent from the genome, but maternal knockdown of *Tc-Dnmt1* leads to embryonic arrest during the first few cleavage cycles of embryogenesis [50]. This indicates a methylation independent role of *Tc-Dnmt1a* in activating the *T. castaneum* egg, which is a much earlier defect than the earliest observable defects in *N. vitripennis*, which occur well after the activation of the zygotic genome.

The *O. fasciatus* and *T. castaneum* phenotypes represent major disruptions of maternal processes that appear to be independent of methylase function of Dnmt1 proteins. Such non-methylation functions of Dnmt orthologs have been suggested in vertebrates as well [51,52]. We cannot exclude that a non-methylating function of *Nv-dnmt1a* exists, and that it could contribute to the complex phenotypes we observed after knocking down *Nv-dnmt1a* with pRNAi. However, the strong correlation of GBM loss and reduced methylated gene expression after *Nv-dnmt1a* knockdown is most easily explained by an important role for the canonical methylase activity of *Nv-dnmt1a* in the embryo.

We believe that our results provide strong evidence that *Nv-dnmt1a*, through its regulation of GBM, is an essential component of the MZT in *N. vitripennis*. We have shown that pRNAi results in incomplete cellularization and gastrulation failure, which is similar to what has been observed when we inhibit zygotic transcription or knockdown essential MZT genes, such as Smaug or Zelda [39]. Importantly, we show that GBM loss is significantly associated with impeded activation of methylated gene expression during zygotic development. Our results further suggest that the indirect effects of *Nv-dnmt1a* pRNAi on non-methylated genes are mediated by both disruption of mRNA degradation and misregulation of transcription from the zygotic genome. These data provide the first evidence that GBM is essential for both the full activation of methylated genes and the subsequent proper regulation of unmethylated genes during the maternal-zygotic transition. The exact mechanisms by which GBM affects transcription, and how disruption of GBM leads to the failure of downstream developmental regulatory networks in *N. vitripennis* will be important questions to address in order to understand the function and evolution of this widespread but poorly understood epigenetic modification.

## Methods

### Live imaging

All embryos were live-imaged as described in Arsala & Lynch 2017 [39]. Briefly, we live-imaged all embryos at 28C for 15 hours on an Olympus BX-80 inverted microscope under 30X silicone immersion and DIC optics.

### Phalloidin staining & transverse sectioning

Control injected and *Nv-dnmt1a* dsRNA (1ug/ul) injected mothers were allowed to lay eggs on hosts for 2 hours at 25C. Freshly laid eggs were collected from hosts and aged on 1% agarose PBS plates at 25C until they were 8–10 hours old, so that all embryos were aged to correspond with the onset of gastrulation. We hand dissected the chorions of fixed embryos. Afterward, the fixed embryos were placed in 1X PBS and stained with AlexaFluor 488 phalloidin at a

1:250 dilution. The stained, fixed embryos were placed in Vectashield with DAPI (VectorLabs H-1200-10) overnight. Finally, the embryos were laid flat on a cover slip and were cut in half along the transverse axis using a razor slotted into an embryo 'guillotine.' The resulting embryo 'halves' were flipped 'up' on the cover slip and imaged on an Andor Revolution WD spinning disc confocal system. The samples were illuminated with 488nm (phalloidin) and 405nm (DAPI) diode laser and Z-stack images were captured using a 30x objective at 1micron increments.

## RNA extraction

Embryos were collected and aged as described above. We designed 3 non-overlapping dsRNA constructs (see S3 Table for primer information) and injected 100 females per biological replicate with either 1μg/mL dsRNA diluted in water or with water as a control. The females were left unmated, so that all resulting progeny were male as the variation in sex and ploidy could confound our analysis. The injected females were allowed to lay eggs that were aged at 25C until they reached the following four time points across the MZT: 0–2 hours to generate a freshly laid egg transcriptome, 3–5 hours to generate an early blastoderm transcriptome, 5–7 hours to generate a middle blastoderm transcriptome, 7–9 hours to generate late blastoderm embryos that are at the onset of gastrulation. Once the embryos reached the time points, we isolated 100ng of total RNA using TRI Reagent for all samples in biological triplicate. Roughly 50 embryos were collected per sample. We confirmed the RIN value for all samples were above 9.0 using a 2100 Agilent Bioanalyzer and immediately prepared 24 libraries for sequencing.

## RNA library preparation and bulk RNA-sequencing

Using the 100ng of total RNA from each sample isolated above, we performed poly-A tail selection using the NEBNext Poly (A) mRNA magnetic isolation module (E7490) and immediately proceeded with library preparation. Libraries were prepared for bulk RNA-sequencing using the NEBNext Ultra Directional RNA Library Prep Kit for Illumina (E7420). The libraries were purified, validated and pooled before sequencing. The libraries were subjected to 100bp paired-end sequencing across two lanes on a HiSeq4000 (S6 Data). Raw data from this analysis can be found in the NCBI BioProject archive with accession number PRJNA701367.

## RNA-seq analysis

To assess sequencing quality, we used FastQC and saw high quality reads across all samples (S6 Data). We trimmed adapters and removed low quality reads using Trim Galore-0.6.0 only using the—paired parameter. We reassessed the trimmed read quality using FastQC and proceeded to map the reads to genome using HISAT2-2.0.5 (hisat2 -p 8—max-intronlen 10000 -q–x). Differential expression analysis was performed using the R package DESeq2 [53]. We filtered out lowly expressed genes (maximum read value of 10) from our analysis.

## DNA extraction and bisulfite library preparation

We extracted and purified genomic DNA from embryos using the QIAamp DNA Micro Kit (Qiagen 56304). 40ng of genomic DNA was bisulfite converted and libraries were prepared at the University of Chicago Genomics Facility using the Accel-NGS Methyl-Seq DNA Library Kit from Swift Biosciences. Raw bisulfite sequencing reads can be found in the NCBI BioProject archive with accession number PRJNA701143.

### Analysis of Bisulfite-sequencing data

CpG sites were determined to methylated using a binomial test [31,32] where the rate of deamination representing the probability of success and total number of reads representing the number of trials [31,32]. P-values were then corrected for multiple testing [54] and CpGs with adjusted P-values of $< 0.05$ were labeled as methylated CpGs. Genes containing at least one methylated CpG were subsequently labeled as methylated genes. Fractional methylation of CpGs was calculated by dividing the number of methylated reads by the total number of reads and the fractional methylation of genes was calculated by taking the average fractional methylation of all CpGs within the gene body. For the differential methylation analysis, only CpG sites that were methylated in at least one control sample were retained similar to previous insect studies [44,55]. The DSS package [56] was used to assess differentially methylated CpGs with RNAi status as the sole predictor of methylation. P-values for each CpG site was adjusted for multiple testing [54] with significance level set at an adjusted P-value of $< 0.1$. Genes with at least one differentially methylated CpG were considered as differentially methylated genes.

## Supporting information

**S1 Fig. 2-dimensional principal components analysis plots.** Circular points correspond to control samples while triangular points correspond to pRNAi samples. Green = eggs, salmon = early blastoderm embryos, purple = middle blastoderm embryos, blue = late blastoderm embryos.
(TIF)

**S2 Fig. Expression levels of *Nv-Dnmt1a* (Dnmt1a), *Nv-Dnmt1b* (LOC100115455), *Nv-Dnmt1c* (LOC100123657) in egg, early blastoderm, middle blastoderm, and late blastoderm samples are shown, in comparison to those of *Nv-Dnmt3* (LOC100114044) which is phylogenetically well separated from Dnmt1 in *Nasonia* and other insects (Li et al. 2018).** Control is labeled in salmon and pRNAi is labeled in teal. *Nv-dnmt1a* is significantly downregulated in pRNAi samples in all stages. *Nv-Dnmt1b*, which depends on *Nv-Dnmt1a* for its gene body methylation, is significantly down-regulated in middle blastoderm embryos. *Nv-Dnmt1c* and *Nv-Dnmt3* is not differentially expressed at any stage between pRNAi and control embryos. ns = not significant and triple asterisks (***) denote an FDR-adjusted p-value $< 0.0001$.
(TIF)

**S3 Fig. MA plots visualizing gene expression in *Nv-dnmt1a* pRNAi and control samples in freshly laid eggs, early blastoderm embryos, middle blastoderm embryos, and late blastoderm embryos.** Red points indicate significantly up-regulated genes, while blue points indicate significantly down-regulated genes in pRNAi relative to control samples. Gray points are genes with no significant (NS) change in expression in pRNAi samples relative to control samples.
(TIF)

**S4 Fig.** Density distributions of (A) raw transcriptome read counts and (B) normalized counts from DESeq2. Data are log2-transformed. Methylated (salmon) and unmethylated (teal) genes exhibit distinctive patterns of expression consistent with previous studies. The density distributions of the raw and normalized counts are highly similar.
(TIF)

**S5 Fig.** Density distributions of raw transcriptome read counts (top row) and normalized counts (bottom row) for all replicates and stages from DESeq2. Data are log2-transformed. (TIF)

**S6 Fig. Violin plots of methylated and unmethylated genes during the late blastoderm stage in control and Nv-dnmt1a pRNAi embryos.** Top row, all differentially expressed genes (All DEGs), middle, all genes, and bottom, all non-differentially expressed genes (All non-DEGs). p values were determined by performing a 2-tailed T-test assuming unequal variance between RNAi and control. (TIF)

**S7 Fig.** Number (top) and proportion (bottom) of methylated and unmethylated genes, in pRNAi samples, control samples or both, that increase or decrease 2-fold across each developmental transition. (TIF)

**S8 Fig. Differentially expressed methylated and unmethylated gene expression in Nv-Dnmt1a pRNAi and control N. vitripennis embryos.** (A) Heatmap of 1769 hierarchically clustered methylated differentially expressed genes (mDEGs) and their expression over time in control and Nv-Dnmt1a pRNAi samples represented as Z-scores (warmer colors indicate higher than average expression, cooler, lower). Blue and green boxes on the heatmap highlight clusters of mDEGs that are of interest and discussed in the text. (B) Heatmap of 2135 hierarchically clustered unmethylated DEGs (uDEGs) and their expression over time in control and Nv-Dnmt1a pRNAi samples represented as Z-scores (as above). Black, blue, and red boxes on the heatmap highlight clusters of uDEGs that are of interest and discussed in the text. (C) Violin plots of 376 mDEGs in Nv-Dnmt1a pRNAi eggs and early blastoderm embryos. These mDEGs are significantly enriched in the early blastoderm Nv-Dnmt1a pRNAi samples relative to control early blastoderm embryos. Both comparisons were subjected to a t-test. ns = not significant (P>0.05), *** P<0.001 by t-test. (D) Violin plots of 213 uDEGs in control and Nv-Dnmt1a pRNAi early and middle blastoderm embryos, corresponding to the black box in (B). *** P<0.001 by t-test. (TIF)

**S9 Fig.** (A) *Nv-smaug* (LOC100118068) and *Nv-zelda* (LOC100118683) expression in egg, early blastoderm, middle blastoderm, and late blastoderm samples. Control is labeled in salmon and pRNAi is labeled in teal. *Nv-smaug* is only significantly up-regulated in the early blastoderm stage, while *Nv-zelda* is not differentially expressed at any stage. (B) Violin plot of *Nv-smaug* fractional methylation. *Nv-zelda* did not have any detectable levels of methylation in control or pRNAi samples and is considered an unmethylated gene (see Additional File 2). (TIF)

**S1 Table. Size Factors used for normalization of RNA-seq libraries.** (TIF)

**S2 Table. Gene Ontology analysis for down-regulated methylated genes in middle and late blastoderm Nv-Dnmt1a RNAi embryos.** (TIF)

**S3 Table. Numbers of differentially expressed genes (DEGs) between control and Nv-dnmt1a pRNAi samples and their methylation status.** (TIF)

**S4 Table. Primer sequences for *Nv-dnmt1a* knockdown.**
(TIF)

**S1 Data. Normalized RNAseq counts and fractional methylation data.**
(XLSX)

**S2 Data. RNAseq differential expression output.**
(XLSX)

**S3 Data. Improperly activated genes in middle blastoderm pRNAi samples.**
(XLSX)

**S4 Data. Nv-Dnmt1a target genes of interest.**
(XLSX)

**S5 Data. Egg DEGs time course.**
(XLSX)

**S6 Data. Additional experimental information and alignment rates for RNAseq.**
(XLSX)

## Acknowledgments

We thank The University of Chicago Genomics Facility (RRID:SCR_019196), for their very valuable assistance with generating the WGBS and RNA-seq data.

## Author Contributions

**Conceptualization:** Deanna Arsala, Xin Wu, Soojin V. Yi, Jeremy A. Lynch.

**Data curation:** Deanna Arsala, Xin Wu.

**Formal analysis:** Deanna Arsala, Xin Wu, Soojin V. Yi, Jeremy A. Lynch.

**Funding acquisition:** Soojin V. Yi, Jeremy A. Lynch.

**Investigation:** Deanna Arsala, Xin Wu.

**Methodology:** Deanna Arsala, Xin Wu, Soojin V. Yi, Jeremy A. Lynch.

**Supervision:** Soojin V. Yi, Jeremy A. Lynch.

**Validation:** Deanna Arsala, Xin Wu.

**Visualization:** Deanna Arsala, Xin Wu, Soojin V. Yi.

**Writing – original draft:** Deanna Arsala, Xin Wu, Soojin V. Yi, Jeremy A. Lynch.

**Writing – review & editing:** Deanna Arsala, Xin Wu, Soojin V. Yi, Jeremy A. Lynch.

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
