## [Decision Letter · Decision Letter 0]

17 Mar 2022

Dear Dr Lynch,

Thank you very much for submitting your Research Article entitled 'Dnmt1a is essential for gene body methylation and the regulation of the zygotic genome in a wasp' to PLOS Genetics.

The manuscript was fully evaluated at the editorial level and by independent peer reviewers. The reviewers appreciated the attention to an important topic but identified some concerns that we ask you address in a revised manuscript

We therefore ask you to modify the manuscript according to the review recommendations. Your revisions should address the specific points made by each reviewer.

[LINK]

Yours sincerely,

Subba Reddy Palli, Ph.D.

Associate Editor

PLOS Genetics

Wendy Bickmore

Section Editor: Epigenetics

PLOS Genetics

Reviewer's Responses to Questions

**Comments to the Authors:**

Reviewer #1: The revised manuscript by Arsala et al., (Dnmt1a is essential for gene body methylation and the regulation of the zygotic genome in a wasp) presents interesting data regarding the role of gene body methylation (GBM) in gene regulation and development. In my view the most important and clear finding of this manuscript is “that after Nv-Dnmt1a pRNAi, the vast majority of methylated genes that are differentially expressed show reduced expression, and that the magnitude of reduced expression directly correlated with the degree of gene body methylation loss.” The authors also do an excellent job considering the impact of loss of GBM on aspects of development and discuss caveats to interpretation of impacts on development a of a broad disruption of gene regulation from their knockdown of DNA methyltransferase 1a (Nv-Dnmt1a). Another interesting aspect of this work is that, as the authors state, their work suggests that the role of GBM is different in different organisms indicating that over evolution the exact role of GBM can be different in different species. The revised manuscript is suitable for acceptance and will be of broad interest in the field.

Reviewer #2: In this revised version of the manuscript, the authors have addressed most of my points. Here I have some minor comments that might help:

1) The 3-D PCA in figure S1 is quite hard to interpret. Since it is in the Supplementary Material (so not much pressure on space), I would display PCA1 vs PCA2 in a 2D plot (and maybe another with PCA1 vs 3 or 2 vs 3 if required), which is way easier to interpret (as a general rule of thumb, 3D plots are quite confusing).

2) In Figure S7 – Please add a hyphen in the figure legend between “Both – methylated” or “pRNAi only – methylated”. I got confused trying to figure out how come some genes were “pRNAi only methylated”, wrongly interpreting it as those genes being only methylated in the pRNAi.

3) Some discussion on dnmt1b and dnmt1c could be interesting, these do not seem to be affected by the RNAi and are highly expressed during all the stages (and as previously shown, not required for development). Maybe those are more divergent sequences?

4) The links to the data are still not open.

Reviewer #3: The authors have successfully answered the majority of my queries (and those of other reviewers). In my opinion the manuscript can be accepted for publication in Plos Gen.

**Have all data underlying the figures and results presented in the manuscript been provided?**

Reviewer #1: Yes

Reviewer #2: Yes

Reviewer #3: Yes

PLOS authors have the option to publish the peer review history of their article (what does this mean?). If published, this will include your full peer review and any attached files.

Reviewer #1: No

Reviewer #2: No

Reviewer #3: No

---

## [Editor Report · Decision Letter 1]

1 Apr 2022

Dear Dr Lynch,

We are pleased to inform you that your manuscript entitled "Dnmt1a is essential for gene body methylation and the regulation of the zygotic genome in a wasp" has been editorially accepted for publication in PLOS Genetics. Congratulations!

Yours sincerely,

Subba Reddy Palli, Ph.D.

Associate Editor

PLOS Genetics

Wendy Bickmore

Section Editor: Epigenetics

PLOS Genetics

Comments from the reviewers (if applicable):

**Data Deposition**

http://datadryad.org/submit?journalID=pgenetics&manu=PGENETICS-D-22-00132R1

**Press Queries**

---

## [Editor Report · Acceptance letter]

22 Apr 2022

PGENETICS-D-22-00132R1 

Dnmt1a is essential for gene body methylation and the regulation of the zygotic genome in a wasp 

Dear Dr Lynch, 

We are pleased to inform you that your manuscript entitled "Dnmt1a is essential for gene body methylation and the regulation of the zygotic genome in a wasp" has been formally accepted for publication in PLOS Genetics! Your manuscript is now with our production department and you will be notified of the publication date in due course.

With kind regards,

Livia Horvath

PLOS Genetics

On behalf of:
